# Comparison of Adsorption Capacity and Removal Efficiency of Strontium by Six Typical Adsorption Materials

**Hu Li [1], Kexue Han [2,3], Jinhua Shang [1], Weihai Cai [2,3], Minghao Pan [2,3], Donghui Xu [2,3], Can Du [2,3] and Rui Zuo [2,3,\*]**

[1]   Jinan Rail Transit Group Co., Ltd., Jinan 250000, China; lihu1007@163.com (H.L.); jngjgcb2016@163.com (J.S.)
[2]   College of Water Sciences, Beijing Normal University, Beijing 100875, China; hkx@mail.bnu.edu.cn (K.H.); 202021470002@mail.bnu.edu.cn (W.C.); 202021470021@mail.bnu.edu.cn (M.P.); 201921470026@mail.bnu.edu.cn (D.X.); 201821470002@mail.bnu.edu.cn (C.D.)
[3]   Engineering Research Center of Groundwater Pollution Control and Remediation, Ministry of Education, Beijing 100875, China
\*   Correspondence: zuo1101@163.com; Tel.: +86-10-58802738

**Abstract:** The rapid development and application of nuclear technology have been accompanied by the production of large amounts of radioactive wastes, of which Sr is a typical nuclide. In this study, six typical materials with strong adsorption properties, namely activated carbon, kaolin, montmorillonite, bentonite, zeolite, and attapulgite, were selected. Their adsorption mechanisms were investigated by analyzing their adsorption isotherms, adsorption kinetics, micromorphologies, element contents, specific surface areas, crystal structures, and functional groups. The results showed that the adsorption efficiency of Sr by the six adsorbents can be ranked as zeolite, bentonite, attapulgite, montmorillonite, activated carbon, and kaolin, among which the maximum adsorption capacity of zeolite was 4.07 mg/g. Based on the adsorption kinetic and thermodynamic fitting results, the adsorption of Sr by zeolites, bentonite and attapulgite is consistent with Langmuir model, the pseudo-first-order and pseudo-second-order model, and the adsorption process of Sr (II) by montmorillonite, activated carbon and kaolinite is consistent with the Freundlich model and corresponds to non-uniform adsorption. The main mechanisms of the six materials are physical adsorption, ion exchange and complexation. In summary, zeolite, bentonite, and attapulgite, especially zeolite, are highly effective for the treatment of radioactive wastewater containing strontium and have great application value in the treatment of radioactive wastes.

**Keywords:** activated carbon; Kaolin; Montmorillonite; Bentonite; Zeolite; Attapulgite; Strontium; adsorption efficiency





## 1. Introduction

Currently, nuclear energy has become an extremely important part of the world's energy system. However, the major concern that is limiting its sustainable development is still the security issue [1–3]. Radionuclides produced by nuclear power plants have high chemical and biological toxicity, of which $^{90}$Sr is the most harmful to the human body [4]. In the application of nuclear energy, improper disposal of radioactive wastes can lead to serious consequences, such as destruction of human cells and tissues, resulting in pathological changes and even death [5]. The continuous development of nuclear technology has led to the incessant accumulation of radioactive wastes. To protect the ecological environment and human health, the safe treatment of nuclear fertilizer is becoming increasingly urgent.

Considering the chemical and biological characteristics of radioactive wastes, the common disposal methods include chemical precipitation, ion exchange, adsorption, evaporation and concentration, electrolysis, and redox methods [6]. The advantages and disadvantages of various disposal methods can be seen in Table 1. As we can see, the redox method is costly, the electrolysis method consumes a large amount of electric energy, and

the chemical precipitation method requires the addition of large amounts of chemicals to the waste liquid to be treated, which is liable to cause secondary pollution [7]. Therefore, the adsorption method is the fastest and most efficient method among the above-mentioned treatment methods. In addition, it has the advantages of no secondary pollution and a flexible adsorption process [8].

**Table 1.** Comparison between different disposal methods.

| Disposal Methods | Advantages | Disadvantages |
|---|---|---|
| Chemical precipitation | Low cost, simple method and proven technology | Low selectivity and poor purification |
| Ion exchange | High selectivity and wide range of applications | Higher cost and large amount of waste generated |
| Evaporation and concentration | High efficiency of removal | High thermal energy consumption and high operating costs |
| Electrolysis | Vulnerability to other factors | Incomplete technical system |
| Redox methods | High selectivity | Costly |

In surface disposal projects, the selection of buffer barrier materials is the key to preventing radioactive waste from diffusing into the environment. The types of backfilling and barrier materials determine the effectiveness of the barrier system [9]. In recent years, in the process of using surface disposal methods to treat radioactive wastes, the barrier materials chosen by countries worldwide include clay, cement, and gypsum [10]. Besides, Chitosan and chitin, as typical biopolymers, are also commonly used as a material for adsorption of nucleophiles, but the low mechanical properties and unfavorable pore properties in terms of low surface area and total pore volume limit their adsorption application [11–13]. In contrast, clay has a large specific surface area, high plasticity and strong adsorption, etc. And clay is widely recognized because it meets most of the conditions for the selection of barrier materials [14]. The commonly used clay minerals include bentonite, attapulgite, kaolin, and montmorillonite [2], which have large adsorption capacities, abundant reserves, and excellent application prospects in the process of radioactive waste treatment [15]). Sytas et al. [16] studied the adsorption characteristics of thermally activated bentonite for U(VI). The results showed that it could successfully adsorb and remove U(VI) in solution under optimized conditions. Krishna et al. [17] used modified bentonite to adsorb $Cr^{6+}$. It was reported that when the pH was 1, the removal effect was the highest. The special pore structure of attapulgite made its adsorption and removal effect excellent [18]. Cui et al. [19] studied the adsorption properties of modified attapulgite to Hg (II) in aqueous solutions. The modified attapulgite had a maximum adsorption capacity of 800 mg $g^{-1}$, and the pH value of 5–9 in the aqueous solutions was the best adsorption range. Kaolin has excellent adsorption capacity. Jiang et al. [20] compared the adsorption properties of aluminum sulfate-modified kaolin and unmodified kaolin for Pb(II) in solution. The results showed that modified kaolin not only has a high adsorption and removal capacity for Pb(II), but also has a good removal effect on other metal ions. Gao et al. [21] took methylene blue as a typical pollutant and studied the adsorption capacity of acid-activated kaolinite to it. It was pointed out that acid-activated kaolinite has an enhanced adsorption capacity to methylene blue. Montmorillonite crystal, which is the main component of montmorillonite, has excellent adsorption removal ability [22]. Wu et al. [23] used Fe-modified montmorillonite to adsorb and remove $Cd^{2+}$ in aqueous solution. It was reported that when the pH value in the solution increased, the adsorption capacity of the modified material to heavy metal ions increased significantly. Activated carbon is commonly used in industrial wastewater treatment [24]. Arulkumar et al. [25] used activated carbon to treat Cr (VI) and the adsorption rate reached 90% under optimal adsorption conditions. Tang et al. [26] processed activated carbon using amino and thiol to remove $Cd^{2+}$ and $Pb^{2+}$ from aqueous solutions, indicating potential mechanisms including physical and chemical interactions.

Zeolite has excellent ion exchange potential owing to its unique structure. Li et al. [10] used nano-zero-valent iron-loaded zeolite to remove $Cd^{2+}$, $Pb^{2+}$, and $As^{3+}$ from soils with maximum adsorption capacities of 48.63 mg g$^{-1}$, 85.37 mg g$^{-1}$, and 11.52 mg g$^{-1}$. Moreover, some synthetic materials are also used in the adsorptive removal of Sr. Neolaka et al., found that the maximum adsorption capacity of 4-VP-co-EGDMA was 4.365 mg/g adsorbent at pH 2, 30 min contact time, under 303 K respectively [27], and the adsorption of Cr by the adsorption medium is selective and the presence of other heavy metal ions in the aqueous solution will affect the adsorption and removal effect of the medium [28]; Janusz found that the adsorption of synthetic hydroxyapatite on Sr conforms to the Freundlich equation, and its dissolution and precipitation occur during the adsorption process [29].

In this study, six typical adsorption materials, namely activated carbon, kaolin, montmorillonite, bentonite, zeolite, and attapulgite, were selected to adsorb a typical Sr nuclide in radioactive waste. The mechanisms of Sr (II) adsorption by the six adsorbents were studied by the differences in their adsorption isotherms, adsorption kinetics, micromorphologies, element contents, specific surface areas, crystal structures, and functional groups. The findings will provide new insights in the proper treatment of radioactive wastes in the future.

## 2. Materials and Methods

### 2.1. Experimental Materials and Instruments

Six types of adsorption materials were used in each group of experiments: activated carbon, kaolin, montmorillonite, bentonite, zeolite, and attapulgite (chemical pure, Sinopharm Chemical Reagent Co., Ltd., Shanghai, China).

The 100 mg L$^{-1}$ Sr (II) solution was diluted by Sr chloride hexahydrate (analytical reagent, Sinopharm Chemical Reagent Co., Ltd., Shanghai, China). The remaining reagents were hydrochloric acid (analytical reagent, Beijing Beihua Fine Chemicals Co., Ltd., Beijing, China) and sodium hydroxide (analytical reagent, Sinopharm Chemical Reagent Co., Ltd., Shanghai, China).

The instruments used in the experiments are provided in Table 2.

**Table 2.** Instruments used in the experiments.

| No. | Equipment Name | Source and Description |
|-----|----------------|------------------------|
| 1 | Centrifuge | LXJ-IIB, Feige |
| 2 | Electronic balance | BSA224S-CW, Sartorius, Göttingen, Germany |
| 3 | Thermostat oscillator | WS20, Wiggens |
| 4 | pH meter | PHS-3C, Leici |
| 5 | Specific surface analyzer | Quadrasorb SI, Quantachrome |
| 6 | Scanning electron microscope | S-4800, Hitachi High Technologies Corporation, Tokyo, Japan |
| 7 | Electronic dispersive spectrometer | S-4800, Hitachi High Technologies Corporation, Tokyo, Japan |
| 8 | X-ray diffractometer | PANalytical, X'Pert PRO MPD |
| 9 | Fourier transform infrared spectrometer | IRAffinity$^{-1}$, SHIMADZU |

### 2.2. Adsorption Experiments

Under the condition of pH 7, the Sr (II) reserve solution was diluted to 20 mg L$^{-1}$ as the target solution. Activated carbon, kaolin, montmorillonite, bentonite, zeolite, and attapulgite were added according to the solid-liquid ratio of 5 g L$^{-1}$. The mixed solutions were shaken at 30 °C and 200 rpm for 5, 10, 30, 60, 120, 240, 360, 720, 1440, 2880, 4320, and 7200 min, respectively. Then, they were centrifuged at 5000 rpm for 5 min and passed through a filter with a pore size of 0.45 μm. Finally, the residual Sr (II) concentration was determined through Inductively Coupled Plasma- Atomic Emission Spectroscopy (ICP-AES), all experiments were set up in triplicate, and measurement results are averaged.

### *2.3. Adsorption Theory*

#### 2.3.1. Isothermal Adsorption Model

The isothermal adsorption model can macroscopically describe the characteristics of the adsorption process, including the adsorption amount and adsorption strength. The common isotherm adsorption models for liquid-solid adsorption are as follows:

Langmuir isothermal adsorption model [30]:

$$\frac{C_e}{Q_e} = \frac{1}{K_L Q_m} + \frac{C_e}{Q_m} \tag{1}$$

where $C_e$ is the equilibrium concentration of Sr in the solution (mg L$^{-1}$); $Q_e$ and $Q_m$ are the equilibrium adsorption capacity (mg g$^{-1}$) and maximum adsorption capacity (mg g$^{-1}$), respectively; and $K_L$ is the separation factor to judge the reaction of the adsorption process.

Freundlich isothermal adsorption model [31]:

$$\lg Q_e = \lg K_F + \frac{1}{n}\lg C_e \tag{2}$$

where $K_F$ is the constant of the adsorption capacity, $n$ is a constant, and $1/n$ is the adsorption strength.

Temkin isothermal adsorption model [32]:

$$Q_e = \frac{RT}{b_T}\ln a_T + \frac{RT}{b_T}\ln C_e \tag{3}$$

where $T$ is the thermodynamic temperature (K), while $R$, $b_T$, and $a_T$ are constants.

#### 2.3.2. Kinetic Model

The usual kinetic models include the following:

Elovich model [23]:

$$Q_t = \frac{1}{a}\ln(ab) + \frac{1}{a}\ln t \tag{4}$$

where $Q_t$ represents the amount (mg g$^{-1}$) of Sr adsorbed at time $t$, $t$ is the reaction time (min), and $a$ and $b$ are constants.

Two-constant model:

$$\ln Q_t = B + A \ln t \tag{5}$$

where $A$ and $B$ are constants.

Pseudo-first-order model [33]:

$$\ln(Q_e - Q_t) = \ln Q_e - k_1 t \tag{6}$$

where $k_1$ is a constant (1 min$^{-1}$).

Pseudo-second-order model [34]:

$$\frac{t}{Q_t} = \frac{1}{k_2 Q_e^2} + \frac{t}{Q_e} \tag{7}$$

where $k_2$ is a constant (g·mg$^{-1}$·min$^{-1}$).

Intra-particle diffusion model [35]:

$$Q_t = \alpha + k t^{\frac{1}{2}} \tag{8}$$

where $\alpha$ (mg g$^{-1}$) and $k$ (mg·g$^{-1}$·min$^{-1/2}$) are constants.

### 2.4. Characterizations and Measurements

The basic chemical composition, surface morphology, and internal structure of the six typical adsorption materials were determined by scanning electron microscopy (SEM), electronic dispersive spectroscopy (EDS), specific surface analysis (by the Brunauer-Emmett-Teller method), X-ray diffraction (XRD), and Fourier transform infrared spectroscopy (FTIR). The characterization results were used to analyze and compare the adsorption efficiencies of the six typical adsorption materials for Sr (II).

## 3. Results and Discussions

### 3.1. Comparison of Adsorption Efficiency of Different Adsorbents

The effect of adsorption time on the adsorption of Sr (II) by the six adsorbents is shown in Figure 1.

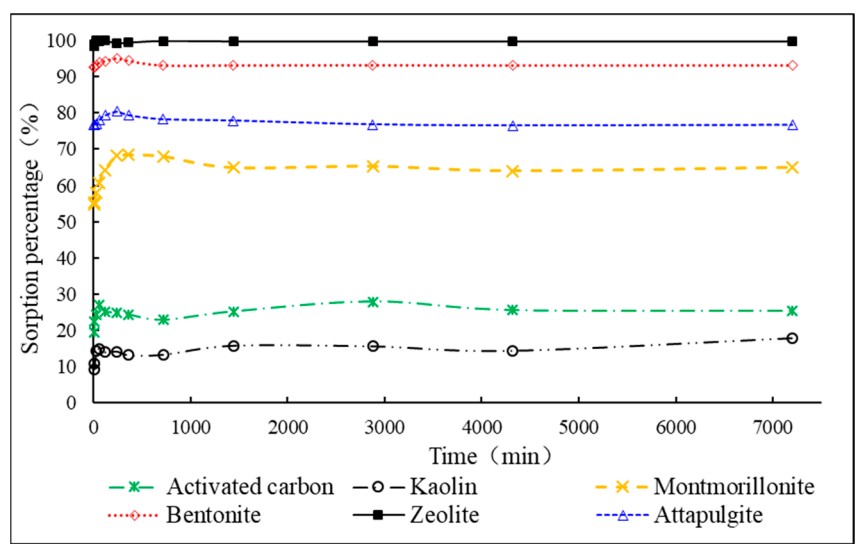

**Figure 1.** Effect of adsorption time on adsorption of Sr (II) by the six adsorbents.

The removal efficiency of Sr (II) by zeolite was the highest (Figure 1). The adsorption efficiency of Sr (II) increased rapidly within 30 min of adsorption, reaching the maximum at 30 min. The rate of adsorption reached 99.9%, and the adsorption capacity reached 4.07 mg g$^{-1}$. The adsorption efficiencies of Sr (II) by bentonite, attapulgite, and montmorillonite were lower than that by zeolite, and the adsorption equilibrium was reached at 360 min after the start of adsorption. The removal rates of Sr (II) by bentonite, attapulgite, and montmorillonite were 95.1%, 80.4%, and 68.4%. The adsorption capacities were 3.85 mg g$^{-1}$, 3.16 mg g$^{-1}$, and 2.78 mg g$^{-1}$, respectively. The adsorption efficiencies of activated carbon and kaolin on Sr (II) were the lowest. The removal rate of Sr (II) by activated carbon was 25.2%, whereas its adsorption capacity was 1.03 mg g$^{-1}$; those of kaolin were 15.8% and 0.63 mg g$^{-1}$, respectively.

Based on the results, the adsorption capacities of different materials for Sr and the comparison of previous studies are shown in Table 3, The difference in the adsorption capacity of the adsorption medium in this experiment and previous studies is due to the fact that the medium used in this experiment is unmodified in any way and is different from the others. According to the experimental results, the adsorption efficiency of zeolite on Sr (II) was the highest. The adsorption rate and capacity of zeolite were nearly four times compared to those of activated carbon. It was followed by bentonite, whose adsorption rate and capacity were nearly five times compared to those of kaolin. Activated carbon and kaolin had the worst adsorption effect.

**Table 3.** The adsorption capacities of different materials for Sr.

| Adsorption Materials | Adsorption Capacities (mg g$^{-1}$) | Contrast Value (mg g$^{-1}$) | References |
|---|---|---|---|
| Activated carbon | 1.03 | 2.5 | [20] |
| Kaolin | 0.63 | 4.2 | [16] |
| Montmorillonite | 2.78 | 15 | [19] |
| Bentonite | 3.85 | 4.5 | [12] |
| Zeolite | 4.07 | 11.52 | [10] |
| Attapulgite | 3.16 | 3.25 | [36] |

### 3.2. Adsorption Isotherms and Kinetics

3.2.1. Adsorption Isotherms

The adsorption isotherms of the six adsorption materials were fitted to compare their differences in the process of Sr (II) adsorption.

Comparing the fitting coefficients $R^2$ of three isothermal adsorption models in the adsorption of Sr by the typical adsorbent materials (Table 4), it was found that the removal process of Sr (II) by activated carbon and kaolin was in accord with the Freundlich model. This indicates that the adsorption behavior of activated carbon and kaolin on Sr (II) corresponded to non-uniform adsorption and chemical adsorption. The $K_F$ value of activated carbon was higher than that of kaolin, indicating that the removal capacity of activated carbon for Sr (II) was better than that of kaolin, which was consistent with the experimental results. The removal of Sr (II) by bentonite, zeolite, and attapulgite accorded with the Langmuir model, indicating that the removal of Sr (II) by the three materials corresponded to monolayer adsorption. However, the $K_L$ value of zeolite was >1, which means that the removal of Sr (II) by zeolite does not easily occur. The adsorption capacity ($Q_m$) of zeolite for Sr (II) was the largest, which was consistent with the experimental results that zeolite had the highest adsorption efficiency. The Freundlich, Langmuir, and Temkin isothermal adsorption models fitted satisfactorily with the adsorption process of Sr (II) by montmorillonite. The parameters indicated that the removal reaction of Sr (II) by montmorillonite is easy.

**Table 4.** Fitting results of isothermal adsorption models for Sr (II) by the six materials.

| Samples | Freundlich Isotherm | | | Langmuir Isotherm | | | Temkin Isotherm | | |
|---|---|---|---|---|---|---|---|---|---|
| | $K_F$ | $1/n$ | $R^2$ | $K_L$ | $Q_m$ | $R^2$ | $A$ | $B$ | $R^2$ |
| Activated carbon | 0.05 | 0.64 | 0.8966 | 0.006 | 2.59 | 0.8511 | 0.39 | 1070.3 | 0.7625 |
| Kaolin | 0.007 | 0.61 | 0.9504 | 0.007 | 1.49 | 0.9511 | 0.19 | 1072.9 | 0.8053 |
| Montmorillonite | 1.08 | 0.38 | 0.9632 | 0.103 | 5.86 | 0.9378 | 1.55 | 2262.3 | 0.9597 |
| Bentonite | 4.83 | 0.42 | 0.6371 | 0.268 | 19.62 | 0.7639 | 2.03 | 532.3 | 0.7810 |
| Zeolite | 48.4 | 0.70 | 0.7745 | 4.002 | 35.67 | 0.8576 | 76.4 | 468.1 | 0.7211 |
| Attapulgite | 1.96 | 0.32 | 0.8311 | 0.167 | 7.53 | 0.8387 | 16.92 | 2672.2 | 0.7295 |

3.2.2. Adsorption Kinetics

Based on the experimental data of Sr (II) adsorption from the solution by the different adsorption materials, the pseudo-first-order model, pseudo-second-order model, Elovich model, two-constant model, and intraparticle diffusion model were fitted to obtain the various kinetic parameters of the adsorption process. The results are presented in Table 5.

The kinetic results show that the removal process of Sr (II) by activated carbon accords with the pseudo-first-order model, and $q_1$ is basically consistent with $q_e$ of the experimental results. The removal reaction of Sr (II) by activated carbon is dominated by diffusion in the initial stage [37]. The removal process of Sr (II) from the solution by montmorillonite is more in line with the pseudo-second-order model. The equilibrium adsorption capacity $q_2$ obtained by fitting is 2.67 mg g$^{-1}$, and the difference between the equilibrium adsorption capacity and the experimental results is less than 5%. This indicates that the removal

reaction of Sr (II) by montmorillonite is mainly chemical adsorption. The Elovich model, two-constant model, and pseudo-first-order model can also fit well the removal process of Sr (II) by montmorillonite. The adsorption of Sr (II) by kaolin, bentonite, zeolite, and attapulgite suits well the two-constant model. The pre-adsorption process of Sr (II) by bentonite matches well the pseudo-first-order model. This stage is physical adsorption. The pseudo-second-order model can better fit the whole process of Sr (II) removal from the solution by bentonite. This shows that the removal reaction of Sr (II) by bentonite corresponds to a combination of physical and chemical adsorption. In addition, the fitting coefficients ($R^2$) of the pseudo-first-order and pseudo-second-order models for zeolite are >0.999. The fitting coefficient of the Elovich model for attapulgite is $R^2 > 0.99$, which indicates that the adsorption process of Sr (II) by the above-mentioned materials is not a simple physical or chemical adsorption process, but a complex one.

**Table 5.** Kinetic parameters of Sr (II) adsorption by the six materials.

| Models | Parameters | Activated Carbon | Kaolin | Montmorillonite | Bentonite | Zeolite | Attapulgite |
|---|---|---|---|---|---|---|---|
| Elovich model | $a$ | 45.68 | 26.88 | 22.13 | 29.51 | 28.31 | 35.77 |
| | $b$ | 6.19 | 248.53 | 1.9 | 5.92 | 1.17 | 3.77 |
| | $R^2$ | 0.9545 | 0.8750 | 0.9828 | 0.9951 | 0.9957 | 0.9956 |
| Two-Constant model | $A$ | 0.02 | 0.07 | 0.02 | 0.008 | 0.008 | 0.008 |
| | $B$ | −0.12 | −1.06 | 0.86 | 1.28 | 1.35 | 1.09 |
| | $R^2$ | 0.9538 | 0.8887 | 0.9822 | 0.9999 | 0.9999 | 0.9999 |
| Pseudo-first-order model | $K_1$ | 0.30 | 0.20 | 0.41 | 0.91 | 0.88 | 0.05 |
| | $q_1$ | 1.02 | 0.55 | 2.63 | 3.80 | 4.06 | 3.19 |
| | $R^2$ | 0.9735 | 0.7312 | 0.9765 | 0.9997 | 0.9999 | 0.1538 |
| Pseudo-second-order model | $K_2$ | 0.81 | 0.63 | 0.43 | 4.12 | 3.77 | 0.18 |
| | $q_2$ | 1.03 | 0.57 | 2.67 | 3.81 | 4.07 | 3.19 |
| | $R^2$ | 0.9681 | 0.7526 | 0.9876 | 0.9997 | 0.9999 | 0.6739 |
| Intra-particle diffusion model | $k$ | 0.004 | 0.005 | 0.009 | 0.011 | 0.012 | 0.009 |
| | $R^2$ | 0.0883 | 0.4854 | 0.0449 | -0.0014 | 0.0008 | 0.0003 |

### 3.3. Characterization

To reveal the reasons for the differences in adsorption efficiencies of the six typical materials, especially on why zeolite is superior to the other five materials, different techniques were selected to characterize the materials. The mechanism was revealed from the aspects of micromorphology, element content, specific surface area, crystal structure, and type of functional group.

### 3.3.1. SEM Image Analysis

Activated carbon had a smooth surface with carbon particles of different sizes and a block structure with sharp edges and corners (Figure 2a). Pongener et al. [6] used SEM to characterize plant-synthesized activated carbon. They found that the outer surface was filled with cavities, and the surface channels could be clearly seen, which was different from the activated carbon used in this study. The SEM image of activated carbon showed that there were fewer cavities and channels, which corroborated the results of its poor removal effect. However, kaolin had a rough surface and a lamellar structure (Figure 2b). The internal pore structure of kaolin is similar to that of activated carbon. The adsorption sites of activated carbon and kaolin on different materials of the same quality were few because of the large pore structure and the connection between the lamellae, which was not close enough. Shaban et al. [38] used natural kaolinite composite materials to adsorb Fe and Mn. The SEM images showed a flaky surface, which was consistent with the micromorphology of the kaolin material used in this study.

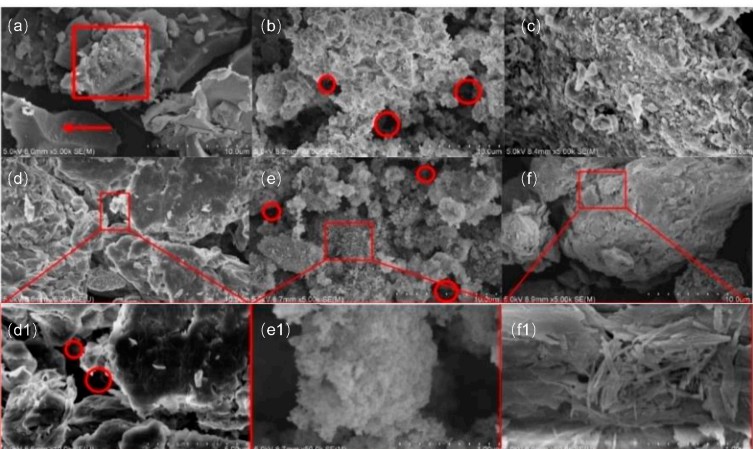

**Figure 2.** SEM images of the six typical materials. (**a**) activated carbon × 5.0 K; (**b**) kaolin × 5.0 K; (**c**) montmorillonite × 5.0 K; (**d**) Bentonite × 5.0 K; (**e**) Zeolite × 5.0 K; (**f**) Attapulgite × 5.0 K; (**d1**) Bentonite × 10.0 K; (**e1**) Zeolite × 50.0 K; (**f1**) Attapulgite × 50.0 K.

Montmorillonite had a massive structure with particles of different sizes, rough surface, thin sheet of sharp needle-like edge, and obvious internal pore development, which was conducive to the adsorption of ions in the solution (Figure 2c). Nguyen-Thanh et al. [39] observed montmorillonite by SEM. The image showed similar characterization results with those in this study. Compared with activated carbon, montmorillonite had a smaller block structure; compared with kaolin, the structure of montmorillonite was tighter and there were more adsorption points. Therefore, the adsorption effect of montmorillonite on Sr (II) is better than those of activated carbon and kaolin. Bentonite was similar to montmorillonite and exhibited a relatively lamellar structure, which could provide a large amount of adsorption space for ions. Compared with the above three materials, bentonite had a superior removal ability because of its well-developed internal channels and suitable internal structure.

Attapulgite had a large amount of rod-like particles and well-developed pores in its microstructure, which enhanced the adsorption ability. The SEM images of original attapulgite soil obtained by Wang et al. [40] showed that attapulgite bundles of rods or fibers (<1 μm) were stacked. The results were consistent with those of Figure 2f.

Zeolite was different from the other adsorbents mentioned above. Its microstructure was composed of many small particles with uneven sizes, forming a non-uniform agglomerate structure and irregular pores. The fine particles increased the adsorption removal capacity of zeolite compared to other materials.

3.3.2. EDS Analysis

The Hitachi S-4800 energy dispersive spectrometer (EDS) was used to determine the composition of elements in the surface of the six typical adsorbents. The energy spectra results of activated carbon, kaolin, montmorillonite, bentonite, zeolite, and attapulgite recorded by EDS are presented in Table 6.

The main components of activated carbon were C and O, while those of kaolin were O and C. The existence of carbon particles in activated carbon made C dominant; thus, the removal effect of activated carbon was better than that of kaolin.

The total content of cationic elements in zeolite (Al, Na, and K), which can exchange with Sr (II) in the solution was 16.68%, while those in bentonite (Mg, Al, Ca, K, and Na), montmorillonite (Mg, Al, and Ca), and attapulgite (Mg, Al, Ca, and K) were 9.93%, 3.53%, and 3.37%, respectively. This indicates that in zeolite, there were significantly more sites provided for Sr (II) to exchange compared to those of bentonite, montmorillonite, and attapulgite in the process of removing Sr (II) from the solution. Therefore, the removal efficiency of Sr (II) from the solution by zeolite was higher than those of bentonite, mont-

morillonite, and attapulgite. Similarly, the removal efficiency of bentonite was better than that of montmorillonite and attapulgite.

**Table 6.** Comparison of element contents in the typical adsorption materials (%).

| Element | Activated Carbon | Kaolin | Montmorillonite | Bentonite | Zeolite | Attapulgite |
|---------|------------------|--------|-----------------|-----------|---------|-------------|
| C | 88.03 | 16.89 | 12.92 | 4.45 | 12.12 | 22.97 |
| O | 11.60 | 62.81 | 66.06 | 67.93 | 58.01 | 58.15 |
| Mg | 0.12 | - | 0.79 | 1.09 | - | 2.24 |
| Al | 0.03 | 10.08 | 2.57 | 7.00 | 7.86 | 3.06 |
| Si | 0.14 | 9.76 | 17.15 | 16.69 | 13.19 | 9.91 |
| Ca | - | - | 0.17 | 0.06 | - | 0.96 |
| Fe | - | 0.06 | 0.34 | - | - | - |
| K | - | - | - | 0.16 | 0.15 | 0.39 |
| Na | - | - | - | 1.62 | 8.67 | - |

Note: "-" means that this element is not included.

However, although attapulgite contained less cationic elements than montmorillonite, the adsorption effect of attapulgite on Sr (II) was better than that of montmorillonite. This is because attapulgite had a unique rod-like structure, which increased its specific surface area to some extent.

### 3.3.3. Specific Surface Area Analysis

The six typical adsorption materials were characterized by Quadrasorb SI surface area analyzer. The results are listed in Table 7.

**Table 7.** Comparison of specific surface areas of the typical adsorption materials.

| Sample | Activated Carbon | Kaolin | Montmorillonite | Bentonite | Zeolite | Attapulgite |
|--------|------------------|--------|-----------------|-----------|---------|-------------|
| Specific surface area (m$^2$/g) | 1407.754 | 10.227 | 183.492 | 28.546 | 110.213 | 205.630 |

Table 5 indicates that the specific area of activated carbon is the largest among the six materials because of its internal pore structure. However, the adsorption rate of Sr (II) by activated carbon was not satisfactory. The reason was the small proportion of the outer surface in the total specific surface compared with the inner surface and the smooth outer surface. The specific surface areas of zeolite and bentonite were smaller than those of attapulgite and montmorillonite, but their adsorption effects were better than those of attapulgite and montmorillonite. The reason is that the total contents of cationic elements in zeolite and bentonite were higher (Section 3.3.2). The cationic contents of montmorillonite were similar to those of attapulgite; however, its specific surface area was smaller than that of attapulgite. Therefore, the adsorption effect of attapulgite on Sr (II) is better than that of montmorillonite. The specific surface area of kaolin was only 10.227 (m$^2$ g$^{-1}$), which was consistent with its worst adsorption effect on Sr (II).

### 3.3.4. XRD Analysis

The XRD images of the six adsorption materials are depicted in Figure 3.

As shown in Figure 3a, activated carbon contained high-purity carbon and silicon dioxide. The XRD image of activated carbon before and after adsorption of Sr (II) had some differences. After adsorption of Sr (II), the diffraction peaks of activated carbon at 2θ = 20.726° and 67.503° disappeared, but those at 2θ = 26.533° and 42.304° weakened. The displacement of the main diffraction peaks at 2θ = 26.533° was not obvious before and after adsorption. This may be due to the adsorption of Sr onto its surface by activated carbon, resulting in a decrease in the intensity of its diffraction peak. After adsorption of Sr (II), the XRD image of bentonite (Figure 3d) showed that a new diffraction peak appeared at

$2\theta = 17.556°$, and the peak at $2\theta = 27.657°$ moved back to $2\theta = 29.041°$. The other diffraction peaks were weakened. The reason for this phenomenon may be the ion exchange reaction and complexation between Sr in bentonite and the minerals in bentonite. The diffraction peaks of kaolin, montmorillonite, zeolite, and attapulgite did not change much.

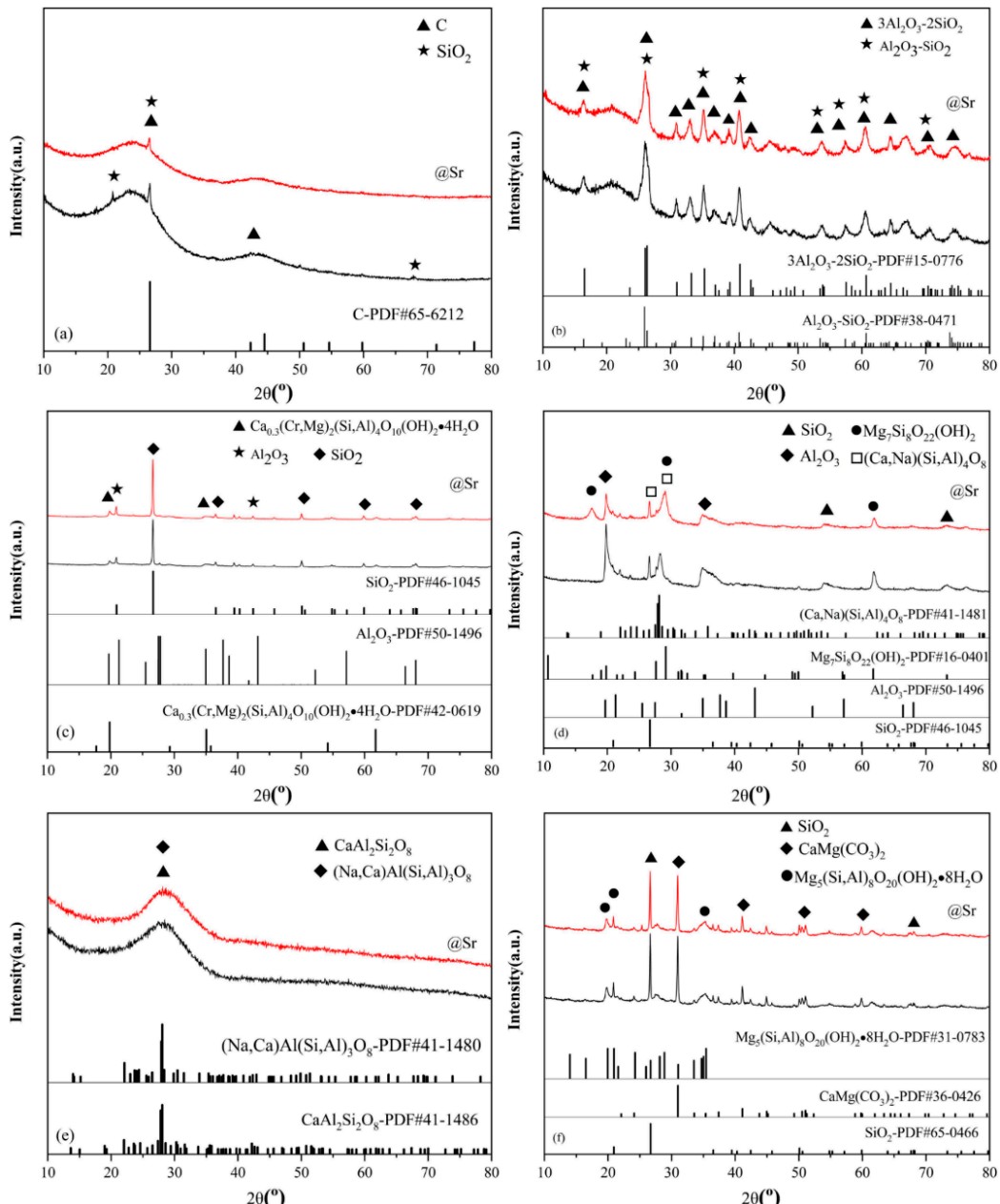

**Figure 3.** XRD images of the six typical materials: (**a**) activated carbon, (**b**) kaolin, (**c**) montmorillonite, (**d**) bentonite, (**e**) zeolite, and (**f**) attapulgite.

According to the XRD results, the change in the characteristic peak strength was the smallest before and after the adsorption of Sr (II) by zeolite, that is, the adsorption of Sr (II) had little effect on the structure of zeolite, which verifies the experimental results that the adsorption capacity of zeolite is the largest among the six materials.

### 3.3.5. FTIR Analysis

The six typical materials were characterized by FTIR, and the results are displayed in Figure 4.

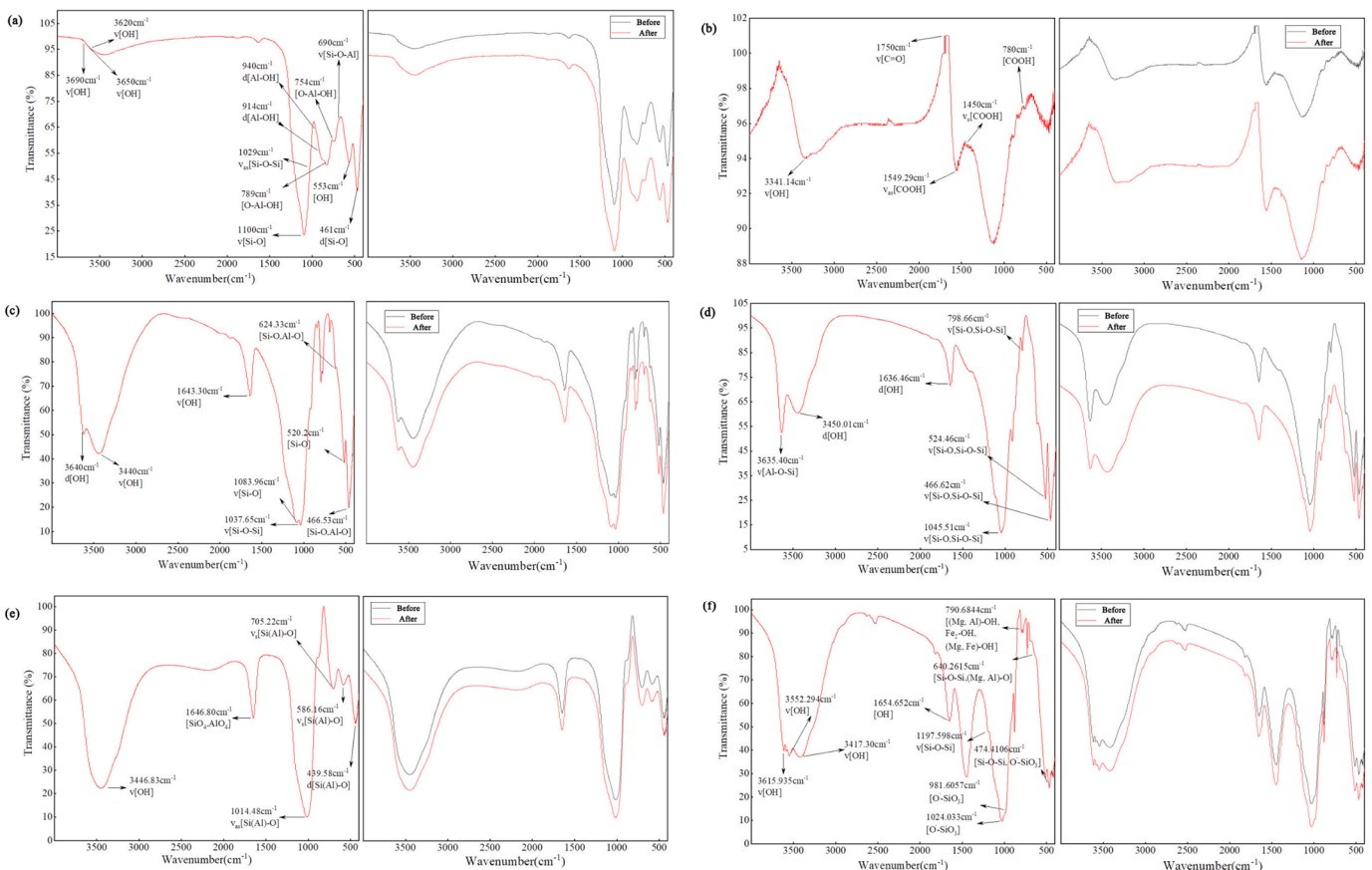

**Figure 4.** FTIR spectra of the six typical materials: (**a**) activated carbon, (**b**) kaolin, (**c**) montmorillonite, (**d**) bentonite, (**e**) zeolite, and (**f**) attapulgite.

In the characterization results of activated carbon, the band at 3340 cm$^{-1}$ is attributed to the O-H stretching vibration. The band at 1750 cm$^{-1}$ is the C=O stretching vibration of the lactone group and carboxyl group, and the bands at 1550 cm$^{-1}$ and 1450 cm$^{-1}$ are attributed to the COOH antisymmetric and symmetrical stretching, respectively. The band at 780 cm$^{-1}$ is attributed to the COO-stretching. It can be observed that activated carbon mainly contains a functional group containing elements of C and O, which is consistent with the EDS spectra analysis results. After adsorption of Sr (II) by activated carbon, the band at 3340 cm$^{-1}$ weakened, which indicates that the bond length of O-H increases or breaks when O binds Sr (II). The other bands did not change significantly. In can be inferred that the adsorption of Sr (II) by activated carbon is a combination of physical and chemical adsorption [41].

The vibration peak of the hydroxyl group on the inner surface of kaolin (-OH) is at 3500 cm$^{-1}$. The band at 1100 cm$^{-1}$ is attributed to Si-O, and that at 940 cm$^{-1}$ is attributed to Al-OH. The bands at 790 cm$^{-1}$ and 690 cm$^{-1}$ are attributed to O-Al-OH and Si-O-Al, respectively. The peak at 461 cm$^{-1}$ is attributed to Si-O [7]. Compared with activated carbon, there are more functional groups containing Si and Al in kaolin, which affect ion exchange in the process of removing Sr (II) from the solution. After adsorption of Sr (II) by kaolin, the adsorption peaks at 1100 cm$^{-1}$, 940 cm$^{-1}$, and 790 cm$^{-1}$ were enhanced. It is indicated that ion exchange adsorption and chemical bonding occurred during the adsorption of Sr (II) [42].

In the high-frequency region, the adsorption band at 3640 cm$^{-1}$ is the bending vibration peak of Al-OH in montmorillonite, and the band at 3440 cm$^{-1}$ is the stretching vibration peak of -OH [43]. In the intermediate frequency region, there is a stretching vibration peak of -OH at 1643.3 cm$^{-1}$, and the bands at 1083.96 cm$^{-1}$ and 1037.65 cm$^{-1}$ are the vibration peaks of the Si-O-Si skeleton. The band at 400–600 cm$^{-1}$ is the internal

vibration of silicon tetrahedron and aluminoxy octahedron [42]. Compared with the FTIR spectra of activated carbon and kaolin, there are more functional groups containing Si in montmorillonite. The results of EDS spectra verify that montmorillonite contains more elements of Si. Among clay minerals, the Si element is the dominant atom for ion exchange in the adsorption process; therefore, the adsorption effect of montmorillonite on Sr (II) in the solution is better than those of activated carbon and kaolin. After adsorption of Sr (II), the FTIR spectra of montmorillonite showed that the strength of -OH stretching vibration peaks at $3640$ cm$^{-1}$ and $3440$ cm$^{-1}$ decreased, and that of Si-O-Si skeleton vibration peaks at $1083.96$ cm$^{-1}$ and $1037.65$ cm$^{-1}$ also decreased. This indicates that ion exchange occurs in the reaction of removing Sr (II).

For bentonite, the stretching vibration peak at $3635.40$ cm$^{-1}$ is related to the Al-O-Si bond, and the bending vibration peaks at $3450.01$ cm$^{-1}$ and $1636.46$ cm$^{-1}$ are related to H-O-H. The bands at $1045.51$ cm$^{-1}$, $798.66$ cm$^{-1}$, $524.46$ cm$^{-1}$, and $466.62$ cm$^{-1}$, are attributed to Si-O-Si, Si-O, and Al-O-Si. After adsorption of Sr (II), the FTIR spectra of bentonite showed that the functional groups in bentonite did not change. The strength of Al-O-Si at $3635.40$ cm$^{-1}$ and Si-O-Si at $1045.51$ cm$^{-1}$ decreased, indicating that ion exchange took place during the adsorption of Sr (II) by bentonite.

In the FTIR spectra of zeolites, the band at $3500$–$3400$ cm$^{-1}$ is the stretching vibration peak of -OH in silicate and aluminate, and the characteristic peak at $1640$ cm$^{-1}$ is the vibration peak of silicon or aluminum oxygen tetrahedron. The bands at $1100$–$1000$ cm$^{-1}$ and $720$–$650$ cm$^{-1}$ are attributed to the antisymmetric stretching vibration and symmetric stretching vibration of the internal tetrahedron of zeolite. The characteristic peak at $440$ cm$^{-1}$ is caused by the bending vibration of Si-O or Al-O. The FTIR spectra of the zeolite after adsorption of Sr (II) showed that the functional groups and peak intensity did not change.

The FTIR spectra of attapulgite are shown in Figure 4f. The bands at $3615$ cm$^{-1}$ and $1654$ cm$^{-1}$ are due to the stretching vibration of OH [44]. The peaks at $1197.598$ cm$^{-1}$ and $640.26$ cm$^{-1}$ are attributed to the stretching vibration of Si-O-Si and O-Mg-O, respectively. The peaks at $1024.033$ cm$^{-1}$ and $474.41$ cm$^{-1}$ are attributed to Si-O with a tetrahedral structure. The FTIR spectra of attapulgite after adsorption of Sr (II) showed that the functional groups and peak strength did not change.

## 4. Conclusions

In conclusion, at a pH of 7, an initial Sr concentration of 20 mg/L, and a temperature of 30 °C, the adsorption efficiencies of the six typical materials for Sr (II) can be ranked as follows: zeolite > bentonite > attapulgite > montmorillonite > activated carbon > kaolin. And zeolite had the strongest adsorption capacity among these six materials, with 4.07 mg/g. Based on the adsorption kinetic and thermodynamic fitting results and combined with the microscopic characterization analysis, the adsorption mechanism of Sr by the six materials can be explained as following: The adsorption of Sr by zeolites, bentonite and attapulgite is consistent with Langmuir model, the pseudo-first-order and pseudo-second-order model, and the adsorption mechanism is mainly physical adsorption and ion exchange. However, zeolite has a larger surface area and higher cation content, so it has a higher adsorption capacity. The adsorption process of Sr (II) by montmorillonite, activated carbon and kaolinite is consistent with the Freundlich model and corresponds to non-uniform adsorption, in which physical adsorption and ion exchange play an important role. It is noteworthy that the functional group species of activated carbon and kaolinite changed significantly before and after adsorption, indicating that complexation has an important effect on their Sr adsorption. In summary, zeolite is preferred as an adsorption material in the treatment of wastewater containing Sr (II), followed by bentonite, attapulgite, and montmorillonite.

**Author Contributions:** Conceptualization, H.L., K.H. and J.S.; methodology, H.L. and J.S.; formal analysis, H.L. and K.H.; investigation, H.L. and W.C.; data curation, H.L., W.C., M.P. and D.X.; writing—original draft preparation, H.L. and K.H.; writing—review and editing, H.L., M.P., D.X. and C.D.; visualization, K.H.; supervision, H.L. and R.Z.; Project Administration, H.L., J.S. and R.Z.; Funding Acquisition, R.Z. All authors have read and agreed to the published version of the manuscript.

**Funding:** This research was funded by the National Key R&D Program of China (No. 2020YFC1806601) and 111 Project (No. B18006).

**Conflicts of Interest:** The authors declare no conflict of interest.

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
