# Peer review of "Comparison of Adsorption Capacity and Removal Efficiency of Strontium by Six Typical Adsorption Materials"

_sustainability, doi:10.3390/su14137723_

Round 1
Reviewer 1 Report
The manuscript with the title "Comparison of adsorption capacity and removal efficiency of strontium by six typical adsorption materials" (sustainability-1760811) is quite interesting. However, to be published in Sustainability, this manuscript needs some improvement. Here are some improvements that need to be considered:
1. The author needs to confirm and clarify the novelty of this research. Is it just about studying the adsorption mechanism of Sr(III) using activated carbon, kaolin, montmorillonite, bentonite, zeolite and attapulgite or is there another novelty aspects for this research?
2. The author needs to explain in more detail the reasons for the importance of selecting Sr(III) for adsorption. This is because there are several radioactive wastes, dyes, heavy metals, pollutants or other samples that need more attention when compared to Sr(III).
3. In the introduction section, the authors are listed several methods which have been developed to remove radioactive wastes (chemical precipitation, ion exchange, adsorption, evaporation and concentration, electrolysis, and redox methods). It is recommended that the authors mentioned the advantage and disadvantages of each technique.
4. The author needs to provide or add reaction mechanism and scheme for adsorption of Sr(III) using activated carbon, kaolin, montmorillonite, bentonite, zeolite and attapulgite. The reaction mechanism and scheme for adsorption of Sr(III) using activated carbon, kaolin, montmorillonite, bentonite, zeolite and attapulgite need to be provided or added based on the results of characterization that has been obtained (FTIR, EDS, SEM, BET and some other characterizations). And as information for the authors, the reaction mechanism and the scheme are different things.
5. The author needs to confirm and clarify why in this research did not study some parameters (other than contact time and heavy metal concentration) that affect adsorption of Sr(III) using activated carbon, kaolin, montmorillonite, bentonite, zeolite and attapulgite, such as initial pH, mass of adsorbents, temperature and some other important parameters. This is considering that some parameters are important to study their effects on adsorption of Sr(III) using activated carbon, kaolin, montmorillonite, bentonite, zeolite and attapulgite, optimization and as reference material for further research. Therefore, the authors need to provide or add a study about the effect of some parameters (other than contact time and heavy metal concentration) that affect adsorption of Sr(III) using activated carbon, kaolin, montmorillonite, bentonite, zeolite and attapulgite.
6. The author needs to confirm and clarify about replication for this research. If this research has been replicated, the author needs to provide or add information about the number of replications that have been carried out. However, if this study has not been replicated, the authors need to replicate the research that has been done. The existence of replication needs to be presented in the form of standard deviation for some Figures and Tables.
7. The author needs to add and determine pH of zero point charge (pHzpc) for activated carbon, kaolin, montmorillonite, bentonite, zeolite and attapulgite. This is considering that information about pH of zero point charge (pHzpc) for activated carbon, kaolin, montmorillonite, bentonite, zeolite and attapulgite is quite important. Here are some papers that can be used as a reference to determine pH of zero point charge (pHzpc) and need to be added in this manuscript:
a. Data in Brief, 16, 2018, 354-360
b. Data in Brief, 16, 2018, 622-629
c. Data in Brief, 17, 2018, 969-979
d. Data in Brief, 17, 2018, 1020-1029
e. Reactive and Functional Polymers, 166, 2021, 105000
f. Journal of Materials Research and Technology, 18, 2022, 2896-2909
8. The author also needs to provide or add and calculate using another error functions (other than R2) for each used adsorption and isotherm models, such as RMSE, SSE, MSE and some other error functions. This is needed to find out which adsorption and isotherm models that can truly represent experimental results. So that the determination of adsorption and isotherm models that can truly represent experimental results is not only seen from the value of R2.
9. The non-linear estimation must be used in the kinetic and isotherm models and also fits and plots
10. The author needs to provide or add studies about selectivity adsorption of Sr(III) onto activated carbon, kaolin, montmorillonite, bentonite, zeolite and attapulgite for binary and ternary mixtures. Here are some papers that can be used as a reference to study selectivity adsorption of Sr(III) onto activated carbon, kaolin, montmorillonite, bentonite, zeolite and attapulgite for binary and ternary mixtures and need to be added in this manuscript:
a. Journal of Environmental Chemical Engineering, 6, 2018, 3436–3443
b. Reactive and Functional Polymers, 147, 2020, 104451
c. Reactive and Functional Polymers, 166, 2021, 105000
11. Desorption study is a very important consideration for the reuse of adsorbents. Therefore, the authors must be performed a desorption experiments and incorporate the results into the manuscript.
12. The author needs to provide or add a comparison between activated carbon, kaolin, montmorillonite, bentonite, zeolite and attapulgite with other materials that have been studied in some previous papers for adsorption of Sr(III). The comparisons that need to be provided or added by the authors can be viewed from several aspects such as economic, environmental, practicality and several other important aspects.
13. The author needs to rearrange the abstract and conclusion according to the comments given by all reviewers
Author Response
With thanks to the comments mentioned by the reviewers, we have responded to each of the comments and have made appropriate changes according to the review comments. The responses are as follows:
- The author needs to confirm and clarify the novelty of this research. Is it just about studying the adsorption mechanism of Sr(II) using activated carbon, kaolin, montmorillonite, bentonite, zeolite and attapulgite or is there another novelty aspects for this research?
Thanks to the reviewers' comments, this paper not only investigates the adsorption characteristics of activated carbon, kaolin, montmorillonite, bentonite, zeolite and attapulgite on Sr, but also compares the six adsorption media by various microscopic characterization methods, thus providing a fundamental theoretical study for the disposal of radioactive nuclear waste containing Sr in nuclear waste disposal sites. The comparison of six adsorption media by various microscopic characterization tools was also carried out, thus providing a basic theoretical study for the disposal of Sr-containing radioactive nuclear waste in geological disposal sites.
- The author needs to explain in more detail the reasons for the importance of selecting Sr(II) for adsorption. This is because there are several radioactive wastes, dyes, heavy metals, pollutants or other samples that need more attention when compared to Sr(II).
As noted in the article,Sr is a typical component in radioactive nuclear waste and has the characteristics of a long half-life and fast migration rate. It is biologically and chemically toxic, easily migrates in environmental media, and accumulates through biological chains. Therefore, it is important to study the sorption behavior of environmental media on the typical nuclide Sr for the prevention and control of Sr contamination.
- In the introduction section, the authors are listed several methods which have been developed to remove radioactive wastes (chemical precipitation, ion exchange, adsorption, evaporation and concentration, electrolysis, and redox methods). It is recommended that the authors mentioned the advantage and disadvantages of each technique.
Thanks for the reviewers’ suggestions, we have compared several treatment methods and analyzed their respective advantages and disadvantages, which can be referred to Table 1.
- The author needs to provide or add reaction mechanism and scheme for adsorption of Sr(II) using activated carbon, kaolin, montmorillonite, bentonite, zeolite and attapulgite. The reaction mechanism and scheme for adsorption of Sr(II) using activated carbon, kaolin, montmorillonite, bentonite, zeolite and attapulgite need to be provided or added based on the results of characterization that has been obtained (FTIR, EDS, SEM, BET and some other characterizations). And as information for the authors, the reaction mechanism and the scheme are different things.
Thanks for the reviewers’ suggestions,In chapter 3.3, we analyzed the characterization changes of adsorption materials before and after adsorption. In the conclusion part, we combined the microscopic characterization with the fitting model to analyze the adsorption mechanism. The adsorption mechanism can be expressed as follows :The adsorption of Sr by zeolites , bentonite and attapulgite is consistent with Langmuir model, the pseudo-first-order and pseudo-second-order model, and the adsorption mechanism is mainly physical adsorption and ion exchange. However, zeolite has a larger surface area and higher cation content, so it has a higher adsorption capacity. The adsorption process of Sr(II) by montmorillonite, activated carbon and kaolinite is consistent with the Freundlich model and corresponds to non-uniform adsorption, in which physical adsorption and ion exchange play an important role. It is noteworthy that the functional group species of activated carbon and kaolinite changed significantly before and after adsorption, indicating that complexation has an important effect on their Sr adsorption.
- The author needs to confirm and clarify why in this research did not study some parameters (other than contact time and heavy metal concentration) that affect adsorption of Sr(II) using activated carbon, kaolin, montmorillonite, bentonite, zeolite and attapulgite, such as initial pH, mass of adsorbents, temperature and some other important parameters. This is considering that some parameters are important to study their effects on adsorption of Sr(II) using activated carbon, kaolin, montmorillonite, bentonite, zeolite and attapulgite, optimization and as reference material for further research. Therefore, the authors need to provide or add a study about the effect of some parameters (other than contact time and heavy metal concentration) that affect adsorption of Sr(II) using activated carbon, kaolin, montmorillonite, bentonite, zeolite and attapulgite.
As the reviewers have commented,the parameters are important to study their effects on adsorption of Sr(II) using activated carbon, kaolin, montmorillonite, bentonite, zeolite and attapulgite,these influencing factors affect the adsorption efficiency of different adsorption media. The aim of this study was to investigate the adsorption and removal effects of different media under the same conditions, and it is regrettable that other experiments could not be carried out. We are looking forward to the relevant research work in our future experiments and will publish the results as soon as possible.
- The author needs to confirm and clarify about replication for this research. If this research has been replicated, the author needs to provide or add information about the number of replications that have been carried out. However, if this study has not been replicated, the authors need to replicate the research that has been done. The existence of replication needs to be presented in the form of standard deviation for some Figures and Tables.
Thanks to the reviewer's comments, we repeat the concentration measurements three times and take the average value each time when conducting experiments
- The author needs to add and determine pH of zero point charge (pHzpc) for activated carbon, kaolin, montmorillonite, bentonite, zeolite and attapulgite. This is considering that information about pH of zero point charge (pHzpc) for activated carbon, kaolin, montmorillonite, bentonite, zeolite and attapulgite is quite important. Here are some papers that can be used as a reference to determine pH of zero point charge (pHzpc) and need to be added in this manuscript:
- Data in Brief, 16, 2018, 354-360
- Data in Brief, 16, 2018, 622-629
- Data in Brief, 17, 2018, 969-979
- Data in Brief, 17, 2018, 1020-1029
- Reactive and Functional Polymers, 166, 2021, 105000
- Journal of Materials Research and Technology, 18, 2022, 2896-2909
Thanks to the reviewers for the comments, we would like to add the pH of zero point charge as follows:
|
Adsorption medium |
pHzpc |
|
activated carbon |
2.5 |
|
kaolin |
4.9 |
|
montmorillonite |
6.0 |
|
bentonite |
7.63 |
|
zeolite |
4.96 |
|
attapulgite |
7.35 |
- The author also needs to provide or add and calculate using another error functions (other than R2) for each used adsorption and isotherm models, such as RMSE, SSE, MSE and some other error functions. This is needed to find out which adsorption and isotherm models that can truly represent experimental results. So that the determination of adsorption and isotherm models that can truly represent experimental results is not only seen from the value of R2.
Thanks to the reviewers for the comments,we regret that we do not have enough data to support our error analysis. We therefore use R2 to determine the fit of the equation, which is also largely plausible, and we will use the error function to make the data more reasonable in future studies
- The non-linear estimation must be used in the kinetic and isotherm models and also fits and plots
Thanks to the reviewers for the comments,we performed adsorption kinetic model fitting and isothermal adsorption fitting based on the experimental data, and the data are shown in Tables 2 and 3. Given the large amount of data and the use of multiple models for fitting in this study, the fitted plots were not put into the manuscript, but this does not affect the accuracy of the experimental data.
- The author needs to provide or add studies about selectivity adsorption of Sr(II) onto activated carbon, kaolin, montmorillonite, bentonite, zeolite and attapulgite for binary and ternary mixtures. Here are some papers that can be used as a reference to study selectivity adsorption of Sr(II) onto activated carbon, kaolin, montmorillonite, bentonite, zeolite and attapulgite for binary and ternary mixtures and need to be added in this manuscript.
- Journal of Environmental Chemical Engineering, 6, 2018, 3436–3443
- Reactive and Functional Polymers, 147, 2020, 104451
- Reactive and Functional Polymers, 166, 2021, 105000
Thanks to the reviewers for the comments, and we have analyzed them in conjunction with the given references.
- Desorption study is a very important consideration for the reuse of adsorbents. Therefore, the authors must be performed a desorption experiments and incorporate the results into the manuscript.
Admittedly, the study of desorption processes is very important for adsorption behavior. Regretfully, we have not conducted any relevant desorption studies, and we will take this aspect into account in our future work.
- The author needs to provide or add a comparison between activated carbon, kaolin, montmorillonite, bentonite, zeolite and attapulgite with other materials that have been studied in some previous papers for adsorption of Sr(II). The comparisons that need to be provided or added by the authors can be viewed from several aspects such as economic, environmental, practicality and several other important aspects.
Thanks to the reviewers for the comments, The purpose of this study is to investigate the adsorption variability and mechanism of Sr adsorption by several materials, and has not yet addressed other aspects of the materials, such as economic and environmental aspects.
- The author needs to rearrange the abstract and conclusion according to the comments given by all reviewers
Thanks to the reviewers for the comments,we have revised and improved it based on the comments made by the reviewers. On this basis, we have reorganized the article and rewritten the abstract and conclusion, and in the conclusion section, we have focused on the analysis of the adsorption behavior of several materials.

Reviewer 2 Report
The presented publication is interesting and a construction is logical. Scientific merit is good. The work is relevant and practical. Clarity of expression and publication of ideas, readability and discussion of concepts is medium. Publication: Comparison of adsorption capacity and removal efficiency of strontium by six typical adsorption materials is very interesting and has practical use. Methodology is well chosen. The authors presented in this the rapid development and application of nuclear technology have been accompanied by the production of large amounts of radioactive wastes, of which Sr is a typical nuclide. In this study, six typical materials with strong adsorption properties, namely activated carbon, kaolin, montmorillonite, bentonite, zeolite, and attapulgite, were selected. Their adsorption mechanisms were investigated by analyzing their adsorption isotherms, adsorption kinetics, micromorphologies, element contents, specific surface areas, crystal structures, and functional groups. The results showed that the adsorption efficiency of Sr by the six adsorbents can be ranked as zeolite, bentonite, attapulgite, montmorillonite, activated carbon, and kaolin. The main mechanisms of Sr adsorption by zeolite, bentonite, and attapulgite were the interaction of monolayer adsorption and various types of adsorptions, including electrostatic interaction, complexation, and ion exchange. The adsorption processes of montmorillonite, activated carbon, and kaolin were in accordance with the Freundlich model and corresponded to non-uniform adsorption. In addition, the adsorption of montmorillonite conformed to the pseudo-second-order model and belonged to chemical adsorption. The adsorption by activated carbon and kaolin matched the pseudo-first-order model and two-constant model, respectively. Characterization results showed that zeolite had the best performance in all aspects; therefore, its adsorption efficiency was the highest. In summary, zeolite, bentonite, and attapulgite, especially zeolite, are highly effective for the treatment of radioactive wastewater containing strontium and have great application value in the treatment of radioactive wastes.
However some corrections are needed:
1. The results in Table 5 should be rounded. It would be good to state how the specific surface area changes after the Sr adsorption process.
2. It would be necessary to unify units according to the Si system.
3. If it is possible, it would be good to do an XRD analysis of the obtained samples, especially before and after adsorption ions and at the same time to calculate the crystallite size.
4. Why the temperature of adsorption 30oC, does it have any application justification?
5. In order to specify the behavior of Sr2+ ions in the system, it would be good to use the MEDUSA program to determine the forms of Sr2+ ions in the tested pH range and concentrations.
6. Papers concerning Comparison of adsorption capacity and removal efficiency of strontium by six typical adsorption materials should be cited in Introduction section; for example:
Hydroxyapatite with magnetic core: Synthesis methods, properties, adsorption and medical applications, Advances in Colloid and Interface Science 291 (2021) 102401
Study of sorption processes of strontium on the synthetic hydroxyapatite Adsorption 22(4-6) (2016) 697-706
Adsorption of the tartrate ions in the hydroxyapatite/aqueous solution of NaCl system Materials 14 (021) 3039

Author Response
With thanks to the comments mentioned by the reviewers, we have responded to each of the comments and have made appropriate changes according to the review comments. The responses are as follows:
- The results in Table 5 should be rounded. It would be good to state how the specific surface area changes after the Sr adsorption process.
Thanks to the reviewers for the comments,it is a pity that we do not have the determination of the specific surface after adsorption. And we will take this aspect into account in our future work.
- It would be necessary to unify units according to the Si system.
Thanks for pointing out the error, we have aligned all units in the manuscript to the SI standard.
- If it is possible, it would be good to do an XRD analysis of the obtained samples, especially before and after adsorption ions and at the same time to calculate the crystallite size.
The XRD changes before and after adsorption have been analyzed, and the changes are more obvious for activated carbon and bentonite, as detailed in 3.3.4. As for the crystallite size, unfortunately, we do not have data on this due to instrumentation.
- Why the temperature of adsorption 30℃, does it have any application justification?
We conducted a pre-experimental study before the formal test, and the data showed that the temperature did not affect the adsorption of Sr by zeolite, and the best adsorption efficiency of activated carbon, kaolin, montmorillonite, bentonite, and attapulgite for Sr was at 30 ℃, so 30 ℃ was used as the experimental temperature.
- In order to specify the behavior of Sr2+ ions in the system, it would be good to use the MEDUSA program to determine the forms of Sr2+ ions in the tested pH range and concentrations.
Comments from reviewers are appreciated,it is preferable to use MEDUSA software to simulate the morphology of Sr in the experimental pH and concentration range. However,in the course of the experiment we kept the initial pH at 7. And under these conditions Sr exists mainly in the form of Sr (II).
- Papers concerning Comparison of adsorption capacity and removal efficiency of strontium by six typical adsorption materials should be cited in Introduction section; for example:
Hydroxyapatite with magnetic core: Synthesis methods, properties, adsorption and medical applications, Advances in Colloid and Interface Science 291 (2021) 102401
Study of sorption processes of strontium on the synthetic hydroxyapatite Adsorption 22(4-6) (2016) 697-706
Adsorption of the tartrate ions in the hydroxyapatite/aqueous solution of NaCl system Materials 14 (021) 3039
Thanks to the reviewers for the comments,the relevant references have been cited in the article.
Reviewer 3 Report
This paper entitled Comparison of adsorption capacity and removal efficiency of strontium by six typical adsorption materials, presents an interesting strategy removal of Sr from waters by various adsorbent materials.
Despite considerable works have been performed several points must be improved for the acceptance of this manuscript.
- Please clearly mention in the last paragraph of the introduction the importance of this work and what the novelty?
- Please explain in the introduction how you chose the series of 6 materials. Why did you not search for biopolymers such as chitin or chitosan for removal of Sr? In the literature, there is a lot of valuable research in this direction. (ex. https://doi.org/10.1016/j.carbpol.2020.116690, https://doi.org/10.1080/01496395.2017.1304961, 10.3390/healthcare7010052)
- Why was the adsorption pH fixed at 7? It would be interesting to conduct researches at lower and higher pH values as well.
- The resolution of Figure 4 is very low. Please improve it.
- Table 2, use the same number of digits and at Kaolin please check the number for two constant model B.
- Please insert a table in order to compare the adsorption capacities for the six used adsorbents with the one presented in the research literature for similar adsorbents.
- The conclusion section must be rewritten. The best optimal conditions, best adsorbent material, and best adsorbent capacity must be presented in this section.
- The references must be adjusted as written in the instructions of Sustainability https://www.mdpi.com/journal/sustainability/instructions
Based on these, I advise the authors to rectify the above mentioned errors and I hope to re-evaluate the revised manuscript.
Author Response
With thanks to the comments mentioned by the reviewers, we have responded to each of the comments and have made appropriate changes according to the review comments. The responses are as follows:
- Please clearly mention in the last paragraph of the introduction the importance of this work and what the novelty?
Thanks to the reviewers' comments, this paper not only investigates the adsorption characteristics of activated carbon, kaolin, montmorillonite, bentonite, zeolite and attapulgite on Sr, but also compares the six adsorption media by various microscopic characterization methods, thus providing a fundamental theoretical study for the disposal of radioactive nuclear waste containing Sr in nuclear waste disposal sites. The comparison of six adsorption media by various microscopic characterization tools was also carried out, thus providing a basic theoretical study for the disposal of Sr-containing radioactive nuclear waste in geological disposal sites.
- Please explain in the introduction how you chose the series of 6 materials. Why did you not search for biopolymers such as chitin or chitosan for removal of Sr? In the literature, there is a lot of valuable research in this direction. (ex. https://doi.org/10.1016/j.carbpol.2020.116690, https://doi.org/10.1080/01496395.2017.1304961, 10.3390/healthcare7010052)
Chitosan and chitin, as typical biopolymers, are also commonly used as a material for adsorption of nucleophiles, but the low mechanical properties and unfavorable pore properties in terms of low surface area and total pore volume limit their adsorption application. In contrast, activated Carbon and clay has a large specific surface area, high plasticity and strong adsorption, etc. and are more suitable as an adsorbent material, so these six materials were selected for this study. This part has been added in the introduction.
- Why was the adsorption pH fixed at 7? It would be interesting to conduct researches at lower and higher pH values as well.
The aim of this study is to investigate the sorption characteristics of Sr by filler materials at geological disposal sites for nuclear waste. Admittedly, the effect of pH on adsorption should not be neglected, but in the actual site the ambient pH is neutral, i.e., around 7, so the pH of the experiment was set to 7.
- The resolution of Figure 4 is very low. Please improve it.
Thanks to the reviewers' comments,we have improved the resolution of Figure 4.
- Table 2, use the same number of digits and at Kaolin please check the number for two constant model B.
We have checked the number for two constant model B,the two are indeed the same.
- Please insert a table in order to compare the adsorption capacities for the six used adsorbents with the one presented in the research literature for similar adsorbents.
Thanks to the reviewers' comments, we have compared the adsorption capacities of the experimental materials in the table.
- The conclusion section must be rewritten. The best optimal conditions, best adsorbent material, and best adsorbent capacity must be presented in this section.
Thanks to the reviewers for the comments,we have presented the best optimal conditions, best adsorbent material, and best adsorbent capacity in this section. On this basis, we have reorganized the article and rewritten the abstract and conclusion, and in the conclusion section, we have focused on the analysis of the adsorption behavior of several materials.
8.The references must be adjusted as written in the instructions of Sustainability https://www.mdpi.com/journal/sustainability/instructions
Thanks to the reviewers' comments, our references are already in the format required by the journal.
Reviewer 4 Report
1. Parameters in table 2 and 3 should be in italics.
2. Sentences in line232-234 can be deleted.
3. In line 354-355, Journal of Hazardous Materials 423 (2022) 127081 could be used as citation for the sepcific confirmation.
4. If possible, please make a clear illustration of adsoprtion of mechanism of Sr on six adsorbents.
Author Response
With thanks to the comments mentioned by the reviewers, we have responded to each of the comments and have made appropriate changes according to the review comments. The responses are as follows:
- Parameters in table 2 and 3 should be in italics.
Thanks for pointing out the errors, we have made the changes and reviewed and revised the content of the full text.
- Sentences in line232-234 can be deleted.
Obviously, this sentence is inappropriate here and we have now removed it.
- In line 354-355, Journal of Hazardous Materials 423 (2022) 127081 could be used as citation for the sepcific confirmation.
We have cited this literature in the manuscript based on the review comments, and the citation here allows for a more scientific description
- If possible, please make a clear illustration of adsoprtion of mechanism of Sr on six adsorbents.
Thanks for the reviewers’ suggestions,In chapter 3.3, we analyzed the characterization changes of adsorption materials before and after adsorption. In the conclusion part, we combined the microscopic characterization with the fitting model to analyze the adsorption mechanism. The adsorption mechanism can be expressed as follows: The adsorption of Sr by zeolites , bentonite and attapulgite is consistent with Langmuir model, the pseudo-first-order and pseudo-second-order model, and the adsorption mechanism is mainly physical adsorption and ion exchange. However, zeolite has a larger surface area and higher cation content, so it has a higher adsorption capacity. The adsorption process of Sr(II) by montmorillo-nite, activated carbon and kaolinite is consistent with the Freundlich model and corre-sponds to non-uniform adsorption, in which physical adsorption and ion exchange play an important role. It is noteworthy that the functional group species of activated carbon and kaolinite changed significantly before and after adsorption, indicating that complexation has an important effect on their Sr adsorption.
Round 2
Reviewer 1 Report
Desorption study is a very important consideration for the reuse of adsorbents. Therefore, the authors must be performed a desorption experiments and incorporate the results into the manuscript.
Author Response
Desorption study is a very important consideration for the reuse of adsorbents. Therefore, the authors must be performed a desorption experiments and incorporate the results into the manuscript.
Thanks to the reviewer for the comments. Unfortunately, we have not conducted the relevant experiments and we are not in a condition to do so now due to the impact of COVID-19, so please understand. And we will take this aspect into account in our future work.
Reviewer 2 Report
Accept in present form.
Author Response
Thanks to the reviewer .The article has been revised appropriately with the reviewer's suggestion, and the quality of the article has been greatly improved.
Reviewer 3 Report
The author has made substantial improvements to this article. The manuscript can be accepted for publication after adding some relevant references for the new inserted paragraph in page 2 lines 79-82.
Author Response
The author has made substantial improvements to this article. The manuscript can be accepted for publication after adding some relevant references for the new inserted paragraph in page 2 lines 79-82.
Thanks to the reviewers for the comments,the relevant references have been cited in the article.